# The Dollar Street Dataset: Images Representing the Geographic and Socioeconomic Diversity of the World

**William A. Gaviria Rojas**[*]
Coactive AI

**Sudnya Diamos**[*]
Coactive AI

**Keertan Ranjan Kini**
Coactive AI

**David Kanter**
MLCommons

**Vijay Janapa Reddi**
Harvard University

**Cody Coleman**
Coactive AI

## Abstract

It is crucial that image datasets for computer vision are representative and contain accurate demographic information to ensure their robustness and fairness, especially for smaller subpopulations. To address this issue, we present Dollar Street—a supervised dataset that contains 38,479 images of everyday household items from homes around the world. This dataset was manually curated and fully labeled [2], including tags for objects (e.g. "toilet," "toothbrush," "stove") and demographic data such as region, country and home monthly income. This dataset includes images from homes with no internet access and incomes as low as $26.99 per month, visually capturing valuable socioeconomic diversity of traditionally under-represented populations. All images and data are licensed under CC-BY, permitting their use in academic and commercial work. Moreover, we show that this dataset can improve the performance of classification tasks for images of household items from lower income homes, addressing a critical need for datasets that combat bias.

## 1 Introduction

Large image datasets built from publicly available web data have made it possible to train deep neural networks and realize powerful computer vision applications such as facial recognition [15], object detection [26], optical character recognition [33] and image segmentation [13]. Seminal image datasets, such as ImageNet [9] and Open Images [21] are commonly used for training foundation models [6] that enable powerful downstream applications via transfer learning [37]. Given the proliferation of these models in both industry and government, it is crucial that image datasets are representative and contain accurate demographic information to ensure their robustness and fairness, especially for smaller low-resource subpopulations where sometimes datasets do not even exist [29].

Work to date has highlighted the challenges in mitigating bias within these datasets. For example, in 2018, researchers demonstrated that commercial gender classification systems misclassified darker-skinned females at substantially higher error rates than lighter males, likely due to these facial recognition algorithms being trained on image datasets composed primarily of lighter-skinned subjects [7]. In other work, researchers assessed ImageNet and Open Images for geo-diversity. They observed that both suffer from amerocentric and eurocentric representation bias that impacts the performance of classification tasks on images from other regions [31, 23]. Beyond implicit bias when sourcing images from the web, existing image datasets also suffer from machine-generated labels which are prone to errors and similar biases [25], and limited licensing further prohibits the legal use of these datasets outside of research and academic institutions. Additionally, crucial socioeconomic

---

[*]Equal contribution. The dataset can be downloaded at `https://mlcommons.org/en/dollar-street`
[2]Thanks to Anna Rosling Rönnlund and the Gapminder team.

36th Conference on Neural Information Processing Systems (NeurIPS 2022) Track on Datasets and Benchmarks.

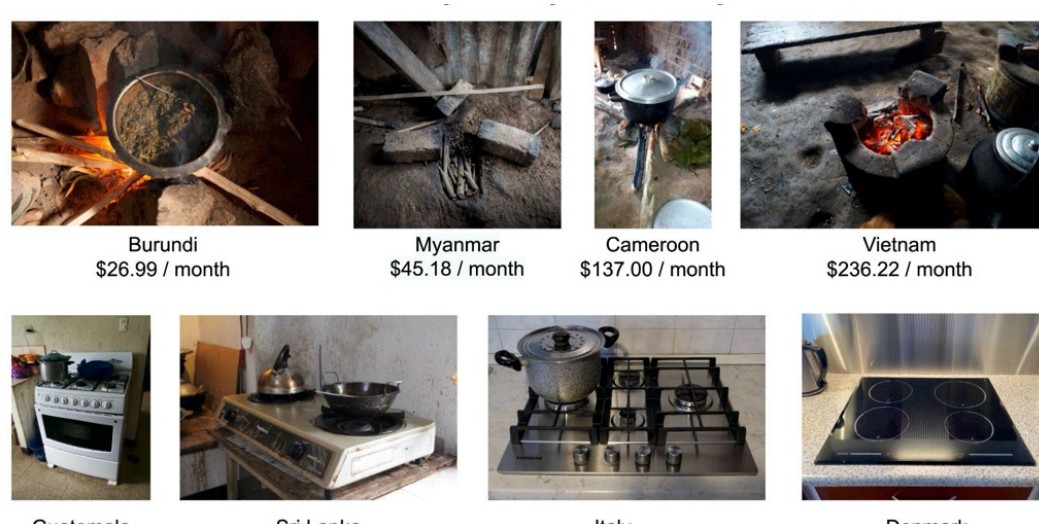

Figure 1: **Pictures of `stove` by country and monthly income in USD.** The Dollar Street dataset contains images of common household items across homes with a wide range of incomes from 63 countries worldwide. As the images in the dataset show, there is a wide discrepancy between what everyday stoves look like across different parts of the world and socioeconomic standing in income.

information (e.g., income, country) is rarely included, and current methods to estimate these from visual information lack often tend to lack generalizability. Devries and Misra [8] are one of the first to examine the accuracy of publicly available object-recognition systems on a geographically diversified dataset. They demonstrate the need to enhance our ML algorithms and datasets so that ML can function equally well for people of various countries where income levels can differ significantly.

In this work, we present the Dollar Street dataset. It is a 38,479 image supervised dataset of 289 everyday household items photographed from 404 homes in 63 countries worldwide. This dataset contains curated images of these objects, appropriate object tags (e.g., "toothbrush", "toilet", "tv") and fully labeled demographic data, including global region, country name and monthly income. Additionally, this data was manually collected and verified, including data from homes with no internet access and incomes as low as $26.99 per month. All data is licensed under the Creative Commons Attribution (CC-BY) license, permitting its use in both academic and commercial work.

Figure 1 shows images of a "stove" across homes of varying locations and incomes. The contrast between a stove in Burundi versus Denmark, for example, reflects the stark socioeconomic differences, demonstrating the diversity captured in this dataset. We also show that when used for training, this dataset can improve the performance of classification tasks by as much as 65%, especially for items from lower-income homes. This highlights the unique snapshot of daily life captured in the Dollar Street dataset and its value in addressing representation and bias in computer vision applications.

The Dollar Street dataset was curated and packaged for the ML research and commercial community by the MLCommons (`mlcommons.org`) organization. To make the dataset more accessible to the research community, we handled the crucial and often challenging tasks of coalescing the raw images from the Dollar Street website into an ML dataset, reviewing the legal implications of the data, filtering non-compliant and sensitive information, and hosting the data. MLCommons will maintain it over time by keeping the dataset up to date as new images become available, handling potential legal issues (e.g., GDPR), permitting the easy withdrawal of content when individuals wish to remove their pictures from the corpus, fixing errors and paying for web hosting of the dataset.

## 2   Related Work

We compare our dataset to relevant published image datasets. Table 1 highlights key differences.

Table 1: Comparison of the Dollar Street dataset with prior related works.

| Dataset name | Images | Classes | Demographic features | License |
|---|---|---|---|---|
| ImageNet | 1.28M | 1K categories | N/A | Non-commercial research and/or educational purposes |
| Open Images | 9.2M | 19.9K categories | Age*, Gender* | CC-BY 2.0 |
| IMDB Wiki Faces | 523K | 20K identities | Age, Gender | Academic research |
| Gender Shades | 1,270 | N/A | Gender, Skin type | Academic research |
| Dollar Street | 38,479 | 289 categories | **Country, Region, Monthly income** | **CC-BY4.0, Commercial** |

*Denotes demographic information with only partial coverage of the dataset.*

**ImageNet** [9, 28] Large Scale Visual Recognition Challenge contains over 1.28M images organized according to the WordNet hierarchy, where each concept in WordNet is called a "synonym set" (abbreviated "synset"). ImageNet contains 1,000 synsets and provides between 732 and 1,300 examples of each synset. Despite widespread impact, the dataset lacks critical demographic features [34, 10] and access is provided for non-commercial research and/or educational purposes only [9].

**Open Images** [21] contains over 9M images annotated with image-level labels for 19,957 categories, as well as object bounding boxes, object segmentation masks, visual relationships and multimodal image descriptions. Machine-generated image-level labels with a substantial false positive rate are provided for all images. In contrast, only a fraction of these images have human-verified labels. The More Inclusive Annotations for People extension [30] provides additional demographic data for gender and race, and the Crowdsourced Extension [14] includes an additional set of images and labels from underrepresented communities. However, both extensions are separate datasets. To the best of our knowledge, no extension exists that combines demographic features and categorical labels.

**IMDB Wiki Faces** [27] contains over 523K face images collected from IMDB and Wikipedia, including gender and age labels. This dataset was created for biological age prediction and is limited to face images of celebrities. Moreover, this dataset was generated from crawling internet data. Its use is limited to academic research due to the lack of appropriate licensing.

**Gender Shades** [7] introduced a dataset containing 1,270 face images of parliamentarians from 6 countries, including their gender and skin type, based on the Fitzpatrick Skin Type classification system. Despite its broader impact, this dataset only includes face images. It is licensed only for academic research, limiting its scope outside of facial analysis benchmarking.

Other works have shown that we can estimate demographic features from image data, such as estimating consumption expenditure and asset wealth from satellite imagery [19], and estimating neighborhood-level demographic attributes (e.g., income, crime rates) from cars detected in Google Street View [12]. While valuable, the derived demographic features can be noisy, and it is dubious whether these methodologies can be generalized to a broader group of images.

## 3 Dataset Description

This section describes how we sourced the raw image data and curated it into a useful ML dataset that represents the geographic and socioeconomic diversity of the world. We briefly describe the contents of the Dollar Street dataset and present the ethical considerations that went into its development.

### 3.1 Contents

The Dollar Street dataset contains 38,479 images collected from homes in 63 countries in four regions (Africa, America, Asia and Europe) as shown in Figure 2. These images capture everyday household items (e.g., "toothbrush," "toilet," "tv") from homes with monthly incomes ranging from $26.99 to $19,671.00 USD. These images are tagged by "topic" tags. Each tag represents one of the 289 household items documented in the photographed homes. The complete topic list is in Appendix A.

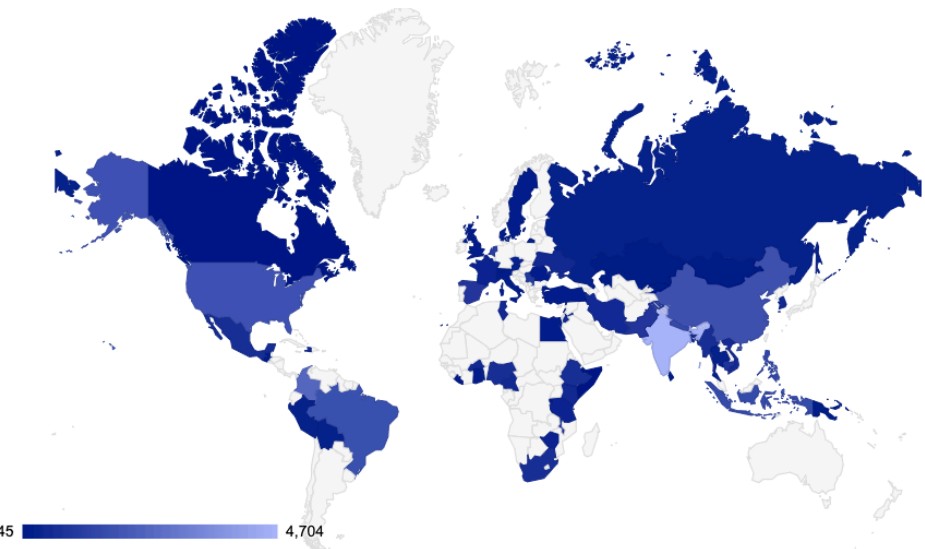

45 ▮▮▮▮ 4,704

Figure 2: **Number of images per country.** The Dollar Street dataset contains contains 38,479 images collected from homes in 63 countries in four regions (Africa, America, Asia and Europe).

The dataset images are 101.3 GB in total size. They are stored in the widely used JPG and PNG formats. Metadata for each image includes a unique identifier, a publicly available URL, an identifier for the home, a set of "topic" tags, and demographic information (i.e., region, country and monthly income). Additional details about the topics are provided in see Section 4.3 and Appendix A.

## 3.2   Licensing

Training machine learning models on public domain work is widely accepted legally. Several datasets built on public domain work already exist and were examined in Section 2. Not having any copyright protections, public domain work allows for unencumbered use for almost any purpose.

Our dataset consists of public domain, CC 4.0 licensed data. CC-BY and CC-BY-SA stand for "Creative Commons Attribution" and "Creative Commons AttributionShareAlike" [11]. We exclude CC-BY-NC-licensed (Creative Commons Attribution-NonCommercial) works because our dataset is intended for downstream commercial and non-commercial usage. Both CC-BY and CC-BY-SA license types allow authors to give content to users to (1) share the work and (2) adapt the work. We interpret the steps to create a machine learning dataset to fit these allowances.

## 3.3   Data Source

The raw data for the Dollar Street dataset comes from the Gapminder foundation—an independent Swedish foundation with no political, religious, or economic affiliations. Images were taken by teams of photographers across 404 homes and 63 countries, with photographers specifically taking images of up to 289 topics per home in the interest of an accurate portrayal of everyday life. Both professional and volunteer photographers collected the photos. The majority of these photographers were discovered by advertising for work on international network websites. In some cases, Gapminder employees approached some photographers once Gapminder discovered their relevant prior work. For example, Gapminder discovered Moa Karlberg's work (https://moakarlberg.com) on food inequality and commissioned her to collect photographs for Dollar Street. Others were sourced via suggestions by coworkers and friends. Each of these photographers spent up to a day in each residence photographing objects such as the family's toothbrushes and favorite shoes. Then, each photograph was tagged by function, family name, country, region and monthly income.

Gapminder used demographic and health surveys for low- and middle-income households around the world to develop a set of parameters for the photographer to employ when searching for families. The photographers were responsible for approaching families directly, frequently with the assistance

of a local "fixer" who helped gather data about the family's consumption, income, and lifestyle. Gapminder went to considerable measures to send professional photographers to disadvantaged and isolated areas, such as a rural Myanmar household, to collect a wide representation of images.

Gapminder provides a raw corpus containing unique image identifiers, URLs to the publicly hosted images and their corresponding metadata. Given our intention to provide this as a ML dataset for academic and commercial use, ensuring the completeness and compliance of the data is important for downstream users. Therefore, we analyzed the metadata for completeness and removed incomplete or non-compliant (see Section 3.4) images and metadata from this raw dataset. We source the images from Gapminder's public website at the highest resolution available from this filtered dataset.

### 3.4 Ethical Considerations

Homes that have been photographed and whose data is included in this dataset have provided informed consent compliant with the time of data collection. Families were informed that "your information is shared to people in many countries, including those outside of the EU and EEA," as detailed in the DollarStreet photo guide that was provided to the photographers [1]. Moreover, they were informed that they have the right to have their images removed from the Dollar Street corpus at any time. All of this was carefully performed through a signed informed consent form that the families signed [5].

Monetary compensation for the families was avoided to ensure participation was completely voluntary and ensure photographers did not and could not "buy" access to homes. Instead, the photographers could offer a symbolic gift, such as a family portrait or a toy for the children, to thank the families for their time and participation. The photographers were reimbursed by Gapminder and they were compensated for their work based on the fair market value of their country.

In the curation process of the Dollar Street data source as a ML dataset, we excluded the images and information for those whose consent predated more recent regulations such as GDPR and are not fully aligned. It is also worth noting that subjects were not compensated for being photographed.

## 4 Dataset Characterization

In this section, we analyze the dataset's images to give some insights into the characteristics of the raw data. We look at the distribution of the images across geography, monthly income and topics. As we demonstrate later in Section 6, this rich diversity of images across countries, regions and income enables us to address the lack of robustness in existing models for socioeconomic differences.

### 4.1 Images by Country and Region

The number of images for each country represented in the dataset is shown in Figure 3. There are 63 countries represented out of 173 that exist worldwide, with the number of images for a given country ranging from 45 (Canada) to 4,704 (India), and a median of 407 images per country.

It is worth noting that the number of images for a given country is not representative of the relative population. For example, despite having an approximately equivalent population to India, China is below India and the United States in coverage. Conversely, Colombia has the second-highest number of images despite its significantly lower population (51 million) than India, the United States and China. Only countries from Africa are not represented in the top 10 countries by the number of images. Malawi is the most represented African country at a rank of 15 with 759 images.

Table 2 highlights the relative distribution of images from different global regions where countries are grouped by continent. The American region combines countries in the North and South American continents. Even though the Asia region has over double the number of images compared with other areas, this region has a significantly higher population (~4.5 B) than others (~1 B). Other regions have approximately the same number of images. It is also worth noting the absence of images from Australia or the broader Oceania region. We will include these in the future when the data is available.

### 4.2 Images by Income

A household's income represents the monthly "consumption" values (in US dollars) of each adult in the home. A family's income is not based solely on wages or raw salary income. The dollars per

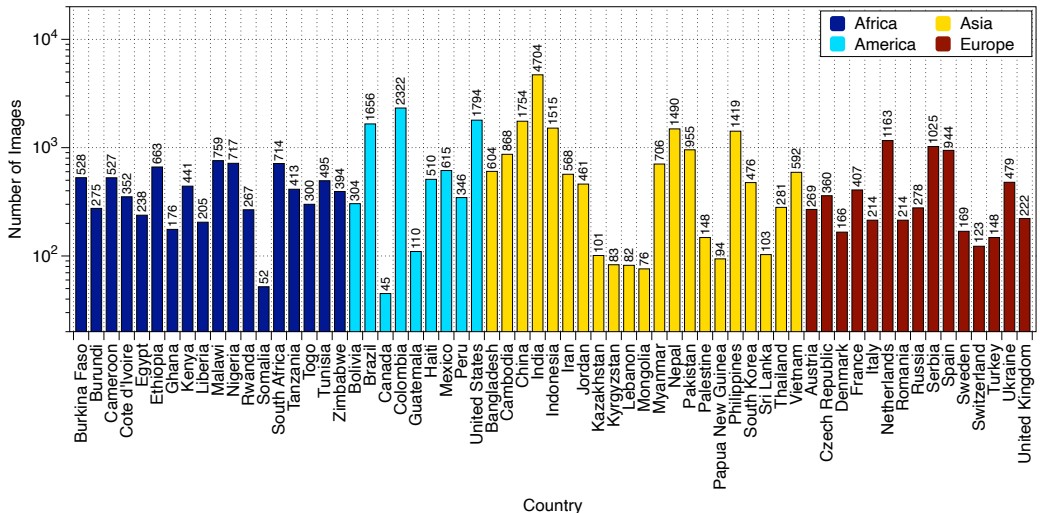

Figure 3: **Number of images per country.** The Dollar Street dataset contains images from 63 different countries around the world. These countries are also spread across four global continents.

month allotted to each household represent the total consumption accessible to each adult inside the household. For instance, if a family grows all of the rice they consume each month, the value of this rice is included in the household's total consumption. This is accomplished by dividing the overall intake by "adult equivalents" to account for the fact that younger children consume less.

Gapminder attempts to quantify consumption as a proxy to estimate the total household income. This number is derived from the self-reported consumption and income levels of the household. The total consumption is measured in U.S. dollars. Gapminder uses dollars adjusted for purchasing power parity (PPP) [20] because the cost of living might vary widely among nations. After sourcing the data, Gapminder examined the official cost of living data for each nation in order to adjust the figures for purchasing power parity (PPP) [20] and converted the results to US dollars [4].

The histogram in Figure 4 highlights the distribution of monthly home incomes for images in the dataset. We observe an approximately similar number of images per bin for monthly incomes ranging up to $25,600 when the binning ranges follow a geometric sequence (i.e. 0-100, 100-200, 200-400, . . . ). Overall, the high variance in monthly income highlights the diversity in GDP per capita for the countries represented in this dataset. It is also worth noting that the median monthly income of $685.00 indicates that the images in this dataset capture valuable visual information about everyday household items that are largely underrepresented in existing image datasets [31].

### 4.3 Images by Topic

Figure 5 shows the number of images for the top 50 topics. As previously noted, each image is tagged from a set of 289 possible topics. We refer the reader to Appendix A for the complete list of topics.

Each image in Figure 5 is tagged with at least one topic. Note, however, that we may tag the same image with multiple topics. For example, for a low-income home where hands are used in place of a toothbrush, a picture of a hand may be tagged as ["hand palm", "toothbrush"]. The most common

Table 2: **Images per region and resolution.** The dataset contains images from four global continents.

|         | Number of Images | Mean     | Median   | Min     | Max      |
| ------- | ---------------- | -------- | -------- | ------- | -------- |
| Africa  | 7516             | 11.51 MP | 14.16 MP | 0.03 MP | 43.74 MP |
| Asia    | 7702             | 11.22 MP | 10.69 MP | 0.01 MP | 42.18 MP |
| America | 17080            | 9.41 MP  | 10.08 MP | 0.02 MP | 64.14 MP |
| Europe  | 6181             | 11.07 MP | 13.31 MP | 0.03 MP | 24.00 MP |

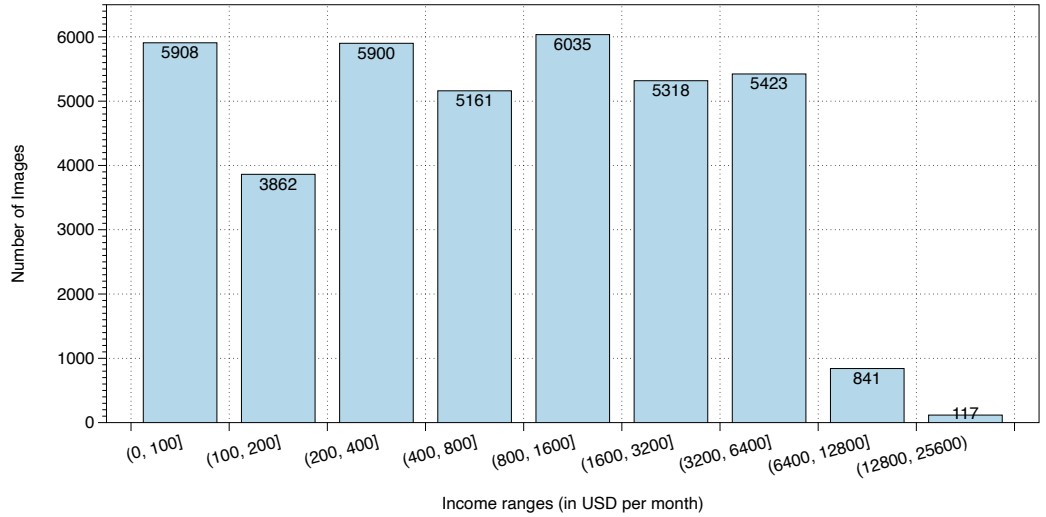

Figure 4: **Number of images by monthly income.** The Dollar Street dataset contains images from homes with monthly incomes ranging from $26.99 to $19,671.00.

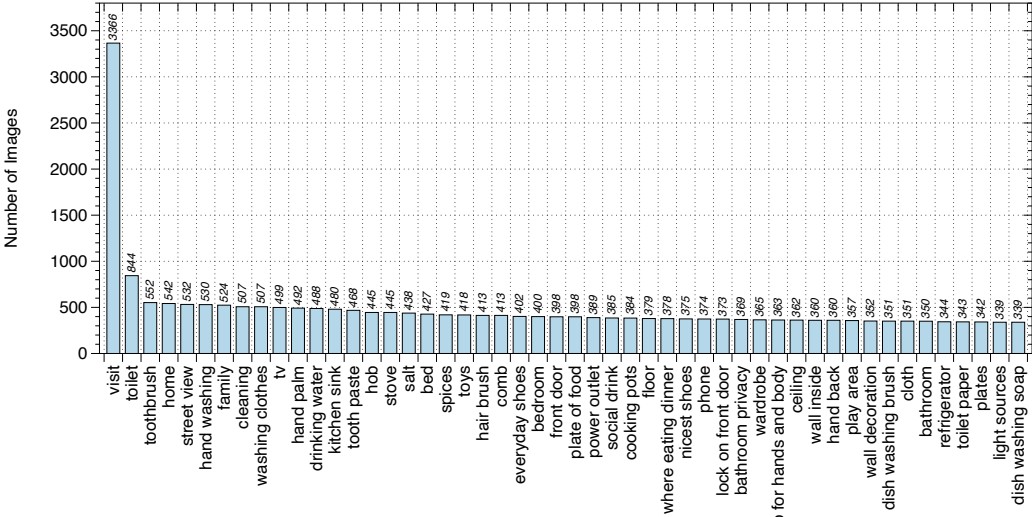

Figure 5: **Number of images by topic.** The Dollar Street dataset contains a total of 289 topics. For brevity we show only the 50 topics here. We have included the complete topic list in Appendix A.

topic tag is "visit." This particular tag denotes images collected by photographers to represent the visited home. The minimum, maximum, and median images by topic are 1, 3366 and 110, respectively. The number of topics per home ranges from 1 to 346, with a median of 118.

## 4.4 Image Characterization

Table 3 highlights the distribution of image resolution in the dataset by quartiles. In aggregate, the images' minimum, maximum, and median resolutions are 6.4 KP, 64.1 MP, and 12.0 MP. Also, in aggregate, the images' minimum, maximum and median image sizes are 2.1 KB, 33.3 MB and 2.4 MB. Across regions, we found that while image resolution and size vary slightly, the mean and other summary statistics were reasonably similar across regions, as shown in the table. The 38,440 and 39 images are stored in JPG and PNG format, respectively. The total size of the images is 101.3 GB.

# 5 Dataset Limitations

It is essential to highlight the detailed definition of the monthly income data [22]. Briefly, monthly income is calculated based on estimated consumption from provided information from the home (i.e., it is not based on the actual dollar income of the home). It refers to the available consumption of each adult in the family. This consumption is then weighted based on age-based consumption, translated to US dollars, and adjusted based on purchasing power parity of the country.

Three additional limitations are discussed in Section 4 and can be briefly summarized here. First, there are approximately twice as many images of homes in Asia compared to any other region, with Indian homes being the most represented in this dataset as shown in Figure 3. Second, the image dataset does not capture homes in Australia or the broader Oceania region. Lastly, approximately half of the images are less than 12 MP in resolution, limiting their impact in some cases, particularly in certain computer vision applications requiring high-resolution imaging.

# 6 Evaluation

To highlight the value of the socioeconomic diversity visually captured in this dataset, we assess the performance of pre-trained models in the visual classification of topics represented in the dataset. We demonstrate that existing models are not robust to socioeconomic differences present in DollarStreet.

## 6.1 Experimental Setup

We use fuzzy matching and manual curation to map 96 topics in our dataset with classes in ImageNet's 1,000 classes [9] (see Appendix B for the exact mapping). We filter out topics with less than 50 images in our dataset and images with no matching ImageNet class. We are left with 21,536 images from the original dataset. We then perform train/test/validation split with a 60:20:20 ratio, ensuring that all images associated with a given family are within a specific split to ensure independence between the splits, validating representativeness. We further split the datasets by quartiles based on monthly income to perform fine-granularity assessments.

We assess the performance of existing pre-trained models from torchvision [3] (i.e. SqueezeNet [18], ShuffleNet [35], ResNet [15], MobileNet [16], DenseNet [17], EfficientNet [32], and VisionNet [36]) on the top-5 multi-class classification accuracy using the mapped ImageNet classes as labels.

To show the significance of the dataset, we also fine-tune the pre-trained ResNet model on all of the training images for 15 epochs and similarly assess its performance after fine-tuning. We used the training schedule from PyTorch's fine-tuning tutorial to avoid extensive hyperparameter tuning [3].

## 6.2 Results

Figure 6 shows the results from the evaluation described above. From the pre-trained models, the average top-5 accuracy across the first, second, third and fourth quartiles were 18.0%, 34.1%, 44.7% and 57.4%, respectively. The correlation between income (shown on the $x$-axis) and model performance (shown on the $y$-axis), particularly the poor performance in classifying items from low-income homes, highlights the importance of the Dollar Street dataset. The dataset can help combat the bias of existing image datasets and foundation models trained on existing datasets.

---

[3]`https://pytorch.org/vision/stable/models.html`

Table 3: Number of images by resolution.

| Resolution in MP | Number of Images |
| --- | --- |
| 0 - 1.85 | 9,633 |
| 1.86 - 12.00 | 9,612 |
| 12.01 - 16.12 | 10,568 |
| 16.13 - 64.14 | 8,666 |

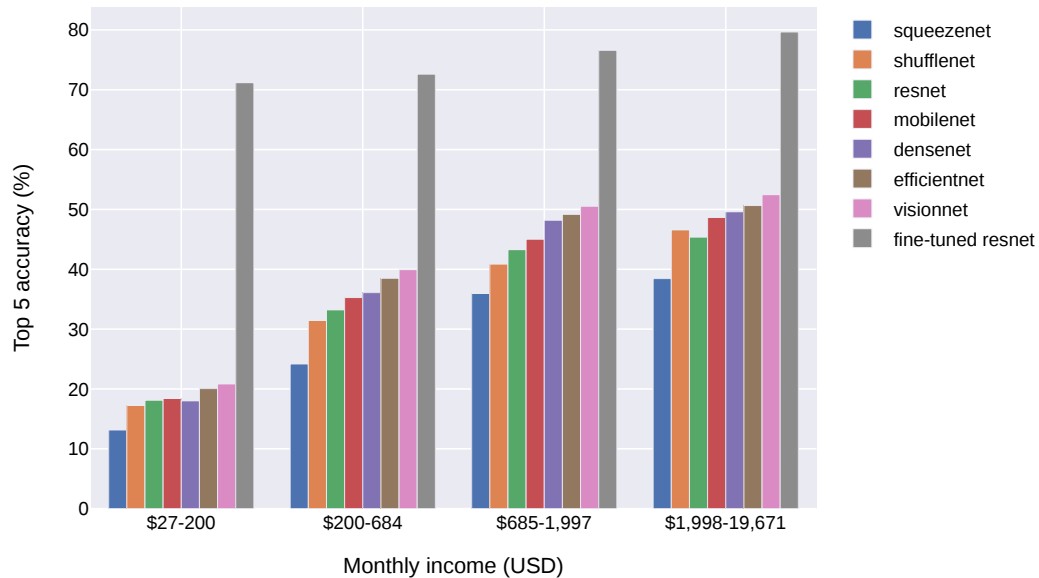

Figure 6: **Top 5 accuracy by monthly income quartile.** The Dollar Street dataset can improve the performance of classification tasks for items from lower-income homes.

Meanwhile, the fine-tuned ResNet model achieved a consistent top 5 accuracy across income quartiles with an average of 75.0%. This is a substantive increase from the top-5 accuracy across the first, second, third and fourth quartiles. This confirms that all the other models are also likely to gain a similar benefit if we use the Dollar Street dataset images on them.

It is worth noting that despite the different model architectures we tested, all pre-trained models show monotonically increasing accuracy with increasing income quartile. We interpret the relatively lower-than-expected performance (even at higher income levels) to signal that the Dollar Street dataset captures a wide visual diversity that is more closely representative of homes across the globe when compared with existing image datasets on which these models were previously trained upon.

### 6.3 Dataset Use Cases

One of the most important applications of this dataset is to promote equity in the market for low-resource image data. This dataset might be used, for instance, to train on photos from a certain geographical region and then test in an unobserved region. It might also be utilized to train on one socioeconomic category and test another. Such application cases will aid in enhancing image datasets for computer vision models. Lessons from the Dollar Street project are transferable to surrounding domains as well. For instance, prior work demonstrated that object recognition-based APIs do not perform well for non-EU or non-North American objects [8].

The users of this dataset should remember that we curated the dataset at a certain point in time. As such, it is a snapshot in time at best. It is important to note that the socioeconomic income of a region is with respect to a certain point in time and conclusions should be drawn accordingly.

Datasets are crucial for machine learning research, but datasets also present us with ethical dilemmas. For instance, the Dollar Street dataset could be used to identify low-resource income communities and intentionally take detrimental actions specifically. The authors do not condone such behavior. We cannot anticipate every conceivable instance of dangerous use. Hence, we request users of this dataset to use their best judgment and adopt sensible and ethical best practices that do no harm to users of downstream tasks that are based on ML models trained using this dataset to serve them.

## 7 Discussion

MLCommons [24], a non-profit organization sponsoring DataPerf(ormance) benchmarking, will sponsor this and future datasets in different domains. As the sponsor and curator, MLCommons will:

- **Update the dataset.** Datasets must continuously evolve to avoid concept drift. To this end, it is essential for public datasets widely used for academic and commercial use to be kept up-to-date. The Dollar Street dataset is based on Gapminder's Dollar Street project [2], enabling us to add new data as it is collected and published by Gapminder. MLCommons commits to keeping Dollar Street updated through periodic revisions.

- **Handle legal issues with the dataset.** It is always possible that a piece of source imagery is in misalignment with the consent requirements under current or future legislation (e.g., GDPR). For example, new permissions may need to be gathered in a changing regulatory environment, or a subject may wish to revoke their consent under new provisions. There needs to be a way for $3^{rd}$ parties to request the removal of such data from the dataset.

- **Permit easy withdrawal of the content.** While Gapminder has collected permission from all participating homes, we will allow anyone to request that images from their home be removed from the dataset. While the Creative Commons license legally permits such use, it is understandable that people may not have given permission with this use case in mind.

- **Fix errors in the dataset and send versioned updates.** Datasets often have problems that are discovered only after their release. When these errors occur in test sets, it becomes less meaningful to make modeling improvements and can limit the value of the dataset.

- **Pay for web hosting of the data.** Hosting large and commercially useful datasets is a costly enterprise. To this end, MLCommons, as the source entity for the dataset, will host the dataset and pay for its hosting fees. The dataset will be available free of charge for its end users. The Dollar Street dataset is 101.3 GB in JPG and PNG format.

MLCommons originally developed this dataset because similar existing datasets generally lack socioeconomic metadata and are not representative of global diversity. MLCommons publishes and hosts this dataset under an CC-BY license to enable wide downstream usage and will maintain it.

## 8 Conclusions

We introduce a new supervised image dataset that contains images of everyday household objects from homes around the world along with the object's function (e.g., "toilet," "toothbrush," "stove"), and the home's country and estimated monthly income (in USD). This image dataset is curated, carefully labeled, socioeconomically diverse, and available commercially, addressing a critical need for datasets that combat bias. We show that this dataset can improve the performance of classification tasks significantly, particularly for images of household items from lower-income homes. To that end, we believe that the Dollar Street dataset is a step forward toward being more inclusive of the rich socioeconomic and geographic diversity that is present in the world for ML computer vision models.

## Acknowledgements

We thank Anna Rosling Rönnlund, Kristin Lagerström, Fernanda Drumond and the team at Gapminder for developing the Dollar Street project and providing their assistance with publishing this dataset. The raw data was provided by Gapminder. The data download, analysis and evaluation were run on Google Cloud Platform instances paid for by Coactive AI. We also thank Zishen Wan from Georgia Institute of Technology for the help with formatting the initial draft of the paper.

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
