# OpenReview forum: "The Dollar Street Dataset: Images Representing the Geographic and Socioeconomic Diversity of the World"
_NeurIPS.cc/2022/Track/Datasets_and_Benchmarks — NeurIPS 2022 Datasets and Benchmarks _

### Official Review · Reviewer_eAoC · 2022-07-20
**Dataset to improve geographic and socioeconomic diversity for 289 categories**

**Rating:** 6
**Confidence:** 3
**Clarity:** Yes, the paper is well written and cl…

**Strengths:**

The strengths of the paper are summarized as follows:

1. The authors address a very relevant issue and provide an intuitive solution. The data is quite diverse, and Figure 1 does an excellent job of depicting the diversity even within a label. The diversity of the dataset could truly help combat inherent bias in machine learning models.
2. The experiments are done well and Figure 6 is a helpful visualization of the model performance to drive home the point that models do better on "higher-income" representations due to the existing bias of the pre-training dataset (ImageNet).
3. The legalities, ethics, and logistics of the dataset are well handled. The authors have a good plan for how the dataset will be maintained.

**Weaknesses:**

The weaknesses of the paper are summarized as follows:

1. The results of the experiment are only presented in Figure 6. It would be more insightful to also see statistical results, and standard deviation.
2. The fine-tuning of ResNet is mentioned but there is no presentation of what hyperparameters were tuned, what ranges were used, and what the final parameters were.
3. Since it is a completely new dataset, I would have appreciated thorough benchmarking of the dataset as well (preliminary results of completely training on the dataset and validation results).

**Additional Feedback:**

1. Is there a timeline as to when Australia and Oceania will be added to the dataset?
2. Since a bulk of the homes are in Asia, is there any plan to balance out the dataset? Any concerns that usage of the dataset can perhaps lead to bias in the other direction?

**Correctness:**

The claims seem to be correct, although it seems like experimental data is missing (specifically, full statistical results). The experiments are sound and well constructed.

**Documentation:**

The dataset has not been released yet. The authors claim they will release it when the paper is accepted. However, there is sufficient detail on collection and on the maintenance plan.

**Ethics:**

The authors address that all residents provided consent on their homes being photographed and that they removed the data of any individuals whose consent predated more recent regulations.

**Relation To Prior Work:**

Yes, the paper goes over existing datasets and explains how they either lack diversity or easy accessibility.

**Summary And Contributions:**

The paper addresses the issue of an existing amerocentric and eurocentric bias in widely used datasets leading to bias in many machine learning applications. So, the authors present DollarStreet, a dataset that of everyday household items from around the world with a variety of countries and incomes. The dataset contains this demographic information as well.

The authors also present experimental analysis on the dataset. After mapping some of the categories to ImageNet and filtering the rest of the dataset, the authors run existing pre-trained torchvision models to show that models do not perform as well, demonstrating the diversity of the dataset compared to ImageNet and other existing datasets. The models also improve in performance with increasing income.

The authors also finetune the pre-trained ResNet model on the training data to show the possible gain in accuracy when used on the dataset.

---

> ### Author Response · Authors · 2022-08-24
> **Additional experiments and details**
>
> We thank the reviewer for their time and thoughtful feedback. Below we have provided detailed responses to each of the weaknesses they raise:
>
> **Statistical results and standard deviation**
>
> We are running the fine-tuning experiments multiple times and will add error bars to Figure 6, but we don’t expect the results to change significantly. The pre-trained models came directly from PyTorch’s torchvision package, and retraining all of them multiple times to create error bars would be too costly for the scope of this work.
>
> **Hyperparameters for fine-tuning ResNet**
>
> We followed the training schedule from PyTorch’s fine-tuning tutorial to avoid extensive hyperparameter tuning and have updated Section 6 with a reference to clarify this.
>
> **Benchmarking the dataset**
>
> We are also working on benchmarking the dataset, but we wanted to be thoughtful about the precedent we set and sanity check (1) how we perform the split and (2) how much data we put in each split. Currently, we are just doing a random split, but there could be a few other options:
>
> 1. Keep it simple and stay with a random split.
> 2. Split based on family id so that all of the images from a family are in one split.
> 3. Create a stratified sample based on the region so that the proportions of regions are maintained across splits.
> 4. Create a stratified sample based on the monthly income quartile so that socioeconomic distribution stays the same across splits.
>
> We reached out to a number of experts and received great feedback already. However, we are still waiting on a few more responses, so we will have to wait and include this in the camera ready.
>
> **Additional feedback on distribution of regions and including Australia and Oceania**
>
> Due to our dependence on Gapminder’s Dollarstreet effort, we are limited by the regions and countries the raw data contained. As Gapminder sources more raw images from more regions and countries, Australia and Oceania may be added, but we cannot give a timeline. The dataset can be supplemented with other sources in the meantime, but we leave that to future work. Our goal with this paper is to release the dataset with as much information as possible and avoid introducing any biases in how the data is interpreted.

---

### Official Review · Reviewer_4aLf · 2022-07-23
**Dollar Street dataset**

**Rating:** 6
**Confidence:** 3

**Strengths:**

A nice addition to this paper is the run benchmark by different image classification models. This shows the potential of this benchmark.

**Weaknesses:**

 The paper does not have any specific weaknesses.


**Additional Feedback:**

As the authors of the paper finetuned some of the models I would assume, that GPU time was used to train those. Therefore I think the GPU point on the checklist is not answered correctly.

**Clarity:**

It was easy readable for me.


**Correctness:**

As mentioned in the weaknesses the benchmark makes for me not so much sense, as the models are all trained on different training data and are evaluated on the test data. This makes a comparison rather difficult. Also the generalisation claim is dubious, as there is no baseline establish, how well the model could fit to the data and we therefore don’t know any discreptance, nor is the test accuracy for the trained models on their data reported.

**Documentation:**

The documentation of the dataset is well done.


**Ethics:**

Missing discussion with sociatal impact. In the paper I could not find any of those dicussions. This would be especially useful, as the dataset could allow algorithms to estimate the income of ones home based on images. This knowledge can of course be potentially harmful and should be mentioned.

**Relation To Prior Work:**

The most important large scale image datasets have been mentioned. Also the distinction between the newly introduced dataset and the previous ones have been made.

**Summary And Contributions:**

The presented paper introduces a new image dataset presenting household items with their respected income value. The dataset contains a new collection of images. Their income tag is estimated by the country income for this demographic group. Afterwards some comparisons with fine tuned networks have been done.

---

> ### Author Response · Authors · 2022-08-25
> **Added training details**
>
> We appreciate the reviewer taking the time to read and evaluate our article. We will update the supplementary material and checklist to include the type of GPU and training time. We also welcome any recommendations you may have for improving the paper. Specifically, we would like to know what the reviewer believes we can do to improve the score (and thereby the technical merits) of our paper.

---

### Official Review · Reviewer_5EDR · 2022-07-24
**Very interesting project attempting to ensure geographic and socioeconomic diversity in everyday household item classification**

**Rating:** 6
**Confidence:** 4
**Correctness:** To the best of my knowledge, the coll…
**Clarity:** This paper is easy to read.

**Strengths:**

**[Diversify the geographic and socioeconomic representation of the data]**

- As an extreme example, the dataset includes images from homes with no internet access and incomes as low as $26.99 per 9 month. These images visually capture valuable socioeconomic diversity of traditionally under-represented populations.

- This dataset can show people from the first world that the data we take for granted has bias. And one way to deal with it is to get data from areas that haven't been represented as much in the past. It demonstrates this with many examples. For example, Figure 1 shows different images of stoves by country and monthly income in USD. It reveals that there is a big difference between how everyday stoves look in different parts of the world and socioeconomic standing in income. Figure 6 vividly demonstrates all pre-trained models show monotonically increasing accuracy with increasing household income.

**[Thoughtful curation]**

- The data is thoughtfully curated and analyzed. Figure 4 shows, for example, that the number of images by monthly income is quite evenly distributed. The authors also provide a reasonable dataset maintenance plan.


**[Commercial License]**

- All images and data are licensed under CC-BY, which allows for their usage in both academic and commercial purposes.  This is in contrast to ImageNet, where access is restricted to non-commercial research and/or educational uses only.


**Weaknesses:**

**[More impacts can be derived from the the value of the socioeconomic diversity captured in this dataset.]**

- I believe the authors of this dataset can claim more impacts. Is the improved performance of classification the only positive thing that happened? If so, how does better classification of objects from different parts of the world affect people's lives? If so, how does the better classification of objects from different places improve people's lives?

- Since this dataset is commercially accessible, it would have been great if the authors provided some practical effect of improved classification of everyday objects.


**[Experiments]**

- > We assess the performance of existing pre-trained models from torchvision on the top-5 multi-class classification accuracy using the mapped ImageNet classes as labels.

    This is the only experiment in the paper, and it is a straightforward way to examine how biased existing pre-trained models are. However, for real change to happen, we need many researchers to add this dataset when they train their models. This way, it can be used by more people and the whole community can benefit from models that are less biased. In this regard, have you done any experiment in reverse? You might, for example, add this dataset to the training data of, say, ImageNet and test on ImageNet evaluation set. My guess is that it will probably hurt the performance. Then, the bigger question will be what kind of effort should we make as a community to encourage people to include this kind of diverse dataset to the evaluation set of widely used benchmarks.


- After manually mapping the classes between ImageNet's 1000 classes and this dataset, you are left with 21K images. About 45 percent of the pictures were gone. I think that matching can be made better or made less strict.



**Additional Feedback:**

None.

**Documentation:**

The authors wrote a lot about where they got the raw image data and how they put it together to prepresent the world's geographic and socioeconomic diversity.

**Ethics:**

No ethical concerns. The dataset has been thoughtfully collected and curated.

Homes that have been photographed and whose data is included in this dataset have provided informed consent compliant with the time of data collection. The authors excluded their images and information for those whose consent predated more recent regulations such as GDPR and are not fully aligned.

**Relation To Prior Work:**

The authors made it clear that none of the relevant large image datasets (ImageNet, OpenImages, etc.) included demographic features such as country, region, monthly income, etc., and there is no extension that combines demographic features and categorical labels.

They also reviewed face datasets for gender, age, skin type, etc. and other works to estimate demographic features from image data.

**Summary And Contributions:**

**[Contributions]**

- The dataset includes 38,479 images of 289 common household objects photographed from 404 homes across 63 nations. Each image has full demographic information such as global region, country name and monthly income.

- The raw data for this dataset comes from the Gapminder foundation—an independent Swedish organization with no political, religious, or economic affiliations.

**[Why]**

- The authors wish to reduce amerocentric and eurocentric representation bias in current datasets (ImageNet, OpenImages, etc.) that hurts the performance of classification tasks on images from other regions.

**[Experiments]**

- When the ResNet model is fine-tuned using this dataset, the performance of classification goes up by up to 50 percent on the evaluation set.

---

> ### Author Response · Authors · 2022-08-24
> **Ideas and opportunities for future work**
>
> We thank the reviewer for their time and thoughtful feedback. Below we have provided detailed responses to each of the weaknesses they raise:
>
> **Making additional claims & practical effect of improved classification**
>
> We appreciate the reviewer's confidence in our work. In addition to the experiments we have undertaken, we included a new section (Section 6.3) that highlights the various ways in which the dataset might be used. However, such concepts are better kept for future development, as they will necessitate extensive further research and rigorous data analysis. This paper focused on the first step of making the dataset accessible to the public, which was a lengthy and arduous procedure to do thoughtfully. Rather than make many additional claims ourselves, we wanted to pave the best possible path for the field so that other researchers could explore the space efficiently.
>
> **Additional experiments combining ImageNet and the DollarStreet Dataset**
>
> We considered running the experiment the reviewer suggested but decided against it given time and resource constraints. Combining the DollarStreet dataset with the ImageNet dataset for training would introduce a number of hyperparameters that would require substantial effort to tune well. First, ImageNet is two orders of magnitude larger than the DollarStreet dataset, so we would need to upsample or heavily weigh the DollarStreet images to compensate. Second, the fuzzy mapping we performed was noisy, as the reviewer mentioned. Tuning both would be a fairly manual process and necessitate extensive research and analysis to draw conclusions.
>
> **Fuzzy mapping with ImageNet**
>
> While we agree that the fuzzy matching was noisy, our goal with the paper is to release the dataset with as much information as possible and avoid introducing biases in how the data is interpreted. The topic labels came directly from Gapminder. We took an initial stab at cleaning up the topics by fuzzy matching them to ImageNet classes, but this was not exhaustive. Defining a concise taxonomy of the topics is important and deserves dedicated effort to avoid introducing subjective biases, which is beyond the scope of this paper. A thorough analysis of this is an excellent opportunity for future work.

---

### Official Review · Reviewer_UKY7 · 2022-07-25
**A sound and convincing dataset that will definitely help addressing socio-economic bias as, e.g., prevalent in current pre-trained image classifiers.**

**Rating:** 10
**Confidence:** 4
**Clarity:** A very well-prepared presentation, ea…

**Strengths:**

The authors definitely delivered a top-notch dataset paper. The addressed problem is clear and of utmost societal relevance, the approach of collecting and tagging large amounts of images showing "basically the same things" across countries of significantly differing economic wealth (and tagging them uniformly) is at the same time creative as well as solid and sound. Details of the dataset such as distributions across countries or items and the method used for income tagging are laid out properly and understandably (also supported by an extensive and well-prepared appendix). This confirms that the dataset is largely well-balanced and serves the goal it is intended for. At the same time, existing (minor) limitations of the dataset are also explained properly.

Trust in the dataset actually serving its goal is also strengthened by the evaluation. Here, the impressively low detection accuracy of widely used pre-trained pytorch image classifiers for low-income images could be vividly confirmed ("all pre-trained models show monotonically increasing accuracy with increasing income quartile." -- what a statement!). At the same time, it is also demonstrated that fine-tuning a pre-trained model with images properly representing the global income spectrum significantly increases detection accuracy for low-income images. This paves the way for less biased and, in the end, less discriminating applications of image classification and recognition in a broad variety of usecases. The dataset thus provides a significant societal value and at the same time allows academia to base and evaluate their respective works on a societally and economically well-balanced and inclusive dataset. The strong support of the MLCommons NGO as well as the collaboration with the well-known Gapminder foundation only adds on top of this.

**Weaknesses:**

The only possible weakness I might see lies in the reliance of a pre-existing dataset. The details of the collection of this underlying dataset are out of the authors' reach and could thus be subject to biases etc. However, the Gapminder foundation sponsoring the initial collection can also be considered scientifically trustworthy (probably even more than it is the case for other image datasets).

Another possible "weakness" regards the extent of the evaluation. In the current form, it is unquestionably sufficient but some even more detailed analyses would be even more interesting.

**Additional Feedback:**

Altogether, the paper at hand excels all NeurIPS dataset papers I have seen so far in matters of societal relevance, originality as well as soundness of the underlying approach, conclusiveness and well-prepared presentation.

If I have not overlooked anything really substantial, this paper *must* be accepted for the datasets and benchmarks track. If it is not, I am willing to fight for it.

**Correctness:**

I did not find any (possible) incorrectness - everything seems sound and solid to me.

**Documentation:**

Yes, even though the dataset repo itself was not publicly available yet (There was, however, the offer to provide limited "preview" access upon request)

**Ethics:**

Nothing beyond what the authors already discussed in the additionally provided material.

**Relation To Prior Work:**

Basically yes, but some more links to other datasets with similar/comparable goals of non-biased representation/inclusiveness would have been nice - maybe even of other kinds than image datasets. Should be easy to extend.

**Summary And Contributions:**

This paper presents the dollar street image dataset - a collection of image data depicting various household items (and several further categories such pictures of whole households or hand palms) typical for and used in countries of significantly differing economic wealth. The paper addresses the well-known problem that widely established machine learning datasets - and especially the ones consisting of images - have a significant bias towards higher-income countries and strongly underrepresent low-income ones.

This imbalance does, for instance, lead to the situation that items such as "toothbrushes", "stoves", or "beds" as broadly used in low-income countries are detected as such by image recognition systems with significantly lower accuracy than those from higher-income countries. Foreseeable implications of such biases materialized in pre-trained models are manifold and vividly discussed in, among others, the FAccT community. In the most simple case, imbalanced dfatasets lead to image recognition software functioning worse in low-income countries (because of toothbrushes not being detected as such, for example). More severe impacts with complete devices not functioning properly are, however, also dicussed and increasingly reported.

The dataset introduced with the paper is explicitly tailored to counteract such imbalances. It is based on a pre-existing image database collected by the Gapminder foundation (and additionally known from the Factfulness book), which was pre-filtered based on formal factors such as licenses that could otherwise hinder the dataset's use in research and industry. In addition, the dataset was extended with additional tags such as income of the represented household, country, region, etc. and packaged as a dataset for ML-based image classification. The dataset is provided under a CC-BY license, providing broad applicability in academia as well as in industry and thereby maximizing the possible impact.

---

> ### Author Response · Authors · 2022-08-24
> **Thank you!**
>
> We thank you for your time and thoughtful feedback. We greatly appreciate your confidence in our work and that you see how valuable the data from the DollarStreet project would be for the NeurIPS and ML community. We also agree that many exciting analyses can be done with this data. To this end, we added a new section (Section 6.3) that discusses the different ways the dataset can be used in addition to our experiments. We welcome suggestions if the reviewer would like to see specific analyses.

---

### Official Review · Reviewer_dcyB · 2022-07-26
**The Dollar Street Dataset: Images Representing the Geographic and Socioeconomic Diversity of the World**

**Rating:** 7
**Confidence:** 4

**Strengths:**

Clarity: This paper was clear and easy to follow for me.

Quality: The main experiment in the paper seemed well done.  A few different models were evaluated including one model fine-tuned on the dataset.

Originality:  The experiments are similar to other experiments in the literature, but the paper goes to great lengths to make the dataset easy to use.  For example, the paper specifies that all images have the right licensing for academic + industrial labs, images will be updated to comply with various standards (e.g., GDPR) and the dataset will be updated with additional images as they are collected.  All of this makes the dataset appealing.

Significance:  The paper both provides a dataset that should be easy for others to use (across both academia and industry) as well as benchmarking a variety of models on image recognition across diverse images.  Besides highlighting an important problem for computer vision models, I could the dataset driving important research in this area.   In particular, one thing the paper did not mention is if they feel it will be easy to add new annotations to the dataset.  If this is the case, then the image data might be a rich source of images for a variety of base tasks (e.g., segmentation, detection, image captioning) similar to how COCO images have been reused for a variety of computer vision tasks.


**Weaknesses:**

Quality:  The main experiment demonstrates that object recognition models will likely fail on data that represent a more diverse view of the world.  However, I had a few issues with the way the main experiment was done:

In Table 3, we see that there are a variety of image resolutions.  Image resolutions/image quality can impact how well a vision model works.  Is image quality correlated with things like geographic region?  This might be an important confounder.
I am somewhat concerned about the fuzzy matching done for labels.  In particular, there are some “noisy” mappings like books → bookcase and get water → water jug.  I would have liked to see more discussion about this.
One experiment that could be run (and partially address the point above) would be to finetune the model on data from one socioeconomic quintile and test on other quintiles with the Dollar Street original labels.  This would “adapt” the vision models to the kinds of labels in the original Dollar Street dataset.
Alternatively, in Devries + Misra et al., they do a human evaluation to see if labels predicted by models are accurate according to human judges.

Though I think the above experiments would definitely strengthen the paper, I don’t question the results as they are in line with others results in this space (e.g., Devries + Misra et al.).

I would have liked to see more example images or have had the chance to examine the images myself to make sure that they looked diverse and high quality.  One concern I have is that the images in Figure 1 do not include any complex scenes which might make the dataset less useful.

Originality:  Others have performed other similar experiments (Devries + Misra et al.).  I see the contribution of this paper is to put together a dataset that can be easily used for this sort of experiment, which is not something done in Devries + Misra et al.

Significance: As mentioned, the experimental results are similar to results obtained in other papers.  My main reason for rating this paper highly is that I believe it will be a rich source of data for the community.

Devries + Misra et al:  (Does Object Recognition Work for Everyone?)[https://arxiv.org/pdf/1906.02659.pdf].


**Additional Feedback:**

Does the dataset include any images of people, and if so are there any concerns sharing/distributing those images?

**Clarity:**

This paper was clear and easy to follow.


**Correctness:**

As far as I can tell, everything is correct.


**Documentation:**

Checklist completed, and supplementary includes a datasheet which I appreciate.  I may have missed it, but I would have liked a link to browse the images myself.

**Ethics:**

Addressed with datasheet, text in paper, and checklist.

**Relation To Prior Work:**

Some other work that should be cited:

A similar idea has already been done, though prior work did not release a dataset in what promises to be an easy to use format:  [Does Object Recognition Work for Everyone?](https://arxiv.org/pdf/1906.02659.pdf])

[Visually Grounded Reasoning across Languages and Cultures](https://arxiv.org/pdf/2109.13238.pdf) shows that language and vision models to not adapt to concepts common in different languages/cultures.


**Summary And Contributions:**

This paper presents the Dollar Street Dataset which includes images with geographic and socioeconomic diversity.  A variety of different image recognition models are evaluated using the dataset and the main finding of the paper is that, without fine-tuning, models do substantially worse on images which are linked to lower socioeconomic status.  After fine-tuning, the gap decreases substantially.

The main contribution of this paper is twofold: release of a new dataset and initial benchmarking across a variety of different image classification datasets.

---

> ### Author Response · Authors · 2022-08-24
> **Additional experiments and analyses**
>
> We thank the reviewer for their time and thoughtful feedback. Below we have provided detailed responses to each of the weaknesses they raise:
>
> **Image resolution and regions**
>
> We found that while image resolution and size vary slightly across regions, the mean and other summary statistics were reasonably similar across regions, as shown below and in Table 3.
>
> ```
> Africa:   mean: 11.51 MB, median: 14.16 MB, min: 0.03 MB, max: 43.74 MB
> Asia:     mean: 11.22 MB, median: 10.69 MB, min: 0.01 MB, max: 42.18 MB
> Americas: mean:  9.41 MB, median: 10.08 MB, min: 0.02 MB, max: 64.14 MB
> Europe:   mean: 11.07 MB, median: 13.31 MB, min: 0.03 MB, max: 24.00 MB
> ```
>
> **Fuzzy matching**
>
> While we agree that the fuzzy matching was noisy, our goal with the paper is to release the dataset with as much information as possible and avoid introducing biases in how the data is interpreted. The topic labels came directly from Gapminder. We took an initial stab at cleaning up the topics by fuzzy matching them to ImageNet classes, but this was not exhaustive. Defining a concise taxonomy of the topics is important and deserves dedicated effort to avoid introducing subjective biases, which is beyond the scope of this paper. A thorough analysis of this is an excellent opportunity for future work.
>
> **Evaluating performance on different quartiles**
>
> We also ran the suggested experiments of fine-tuning and evaluating different socioeconomic quartiles. We found that socioeconomic status still impacts model accuracy. Specifically, quartiles improve the most from fine-tuning on neighboring quartiles. Additional data is included in Figure 1 in the supplementary material under “Additional Data.”
>
> **Relation to prior work**
>
> We updated our references to the related work mentioned by the reviewer, crediting them for revealing the need to improve our ML algorithms and datasets so that ML can function equally well for people of various countries where income levels can differ significantly. In comparison, our focus is on categorization, whereas the work by Devries and Misra et al. is on recognition and the work by Liu and Bugliarello et al. is on image captioning. As the reviewer notes, our work complements this previous work; yet, both studies aim to improve image datasets to benefit machine learning models. In addition, we curated the raw photos into a dataset, which was a lengthy procedure. Therefore, when we make the dataset available to the public, others will be able to repeat our works and expand upon them.
>
> **Complex scenes**
>
> Due to our dependence on Gapminder, we are limited in what scenes we can provide. The goal of Gapminder’s DollarStreet effort is to focus on individual items or objects to demonstrate the diversity, rather than including complex behaviors or patterns that are difficult to grasp. Images enable us to bring to light facts that are easier to comprehend and are more readily available to the general public. This is in contrast to the case with raw data, which can be intimidating and off-putting, which prevents people from looking at what the facts truly indicate.
>
> **Images of people**
>
> Yes, there are images of people in categories such as “family,” but we do not have any concerns. Informed consent was gathered and the images are already publically available at https://www.gapminder.org/dollar-street

---

> > ### Comment · Reviewer_dcyB · 2022-09-02
> > **Thanks!**
> >
> > Thanks for your thorough explanations.  I am happy to keep my rating at 7.

---

### Official Review · Reviewer_6Foc · 2022-07-26
**An important and useful dataset, although the paper could use a little more detail**

**Rating:** 7
**Confidence:** 4

**Strengths:**

The inclusion of lower-income households and geographic regions not typically represented in large image datasets is the biggest strength of this work. These populations, despite their under-representation in the training and deployment process, are subjected to computer vision algorithms. Therefore, it is an important contribution of this work that the algorithms can now be checked for their biases along these dimensions.

A strength that perhaps goes beyond the NeurIPS community and into some social science fields is that some tags used for the photos (often objects, such as "stove") include things like "favorite home decoration" and "most loved toy". Similarly, closer to the NeurIPS goals, is choosing to name certain tags in vague enough ways that it encompasses what both ends of the income spectrum may look like, e.g., "place where eating dinner" [sic].

The income levels are not raw numbers but instead take into account various things, including the purchasing power parity of the country, 'available consumption' of each adult in the family, and others, and is translated to US dollars for easy comparison.

**Weaknesses:**

The lack of standardization in the photos (regarding angle, lighting, etc), potentially very few examples of a single type of an object, and the - relatively - small number of images in the dataset may not make this a very suitable dataset for training machine learning algorithms. However, it would be useful in testing (auditing) them for their geographic and income-related biases as described earlier, so I would encourage the writers to emphasize the latter in their work. This is especially important as the paper very rarely discusses the future use of the dataset beyond a little during the introduction and conclusion.

There are some repetitions in the list of Topics, such as "cooking" / "cooking food" / "preparing food" or "bread - ready" / "bread ready". It is said in the paper that a single image has "a set of topic tags", implying that these very similar tags may apply to the same image, in which case it is unclear why they were not merged. The topics are not described, and while most topics do have self-explanatory labels, the boundaries of the category would need to be implicitly gathered from looking at the images under each similar topic and may be understood differently by different people.

Perhaps due to their focus on improving the representation of certain countries/regions, the dataset does not equally represent all regions/countries.

While it is a strength that the monthly income value is not a raw number to be compared across households, and instead standardized for comparability, this process is not described in detail (the only description is vague: lines 170-174). It would be much better if the process could be elaborated on and the reference explicitly mentioned to be a longer description of the calculation.

There is not a lot of information about how the raw data was collected for the dataset - while it is understandable that the authors are not the same team which collected the raw data, the collection process needs to be described as well (or a description shared elsewhere, referred to, as with the income calculation).

**Additional Feedback:**

My understanding of the experiments (i.e. Section 6) is limited as I do not have a technical background. I would advise other reviewers to be taken into account when considering this section.

**Clarity:**

The paper is well-written, as the descriptions are often clear and simple although specific.

**Correctness:**

The dataset construction seems sound, although there is not enough information about the collection of the images. The evaluation appears to be correctly run, however due to my lack of technical knowledge I cannot verify that entirely.

**Documentation:**

The authors describe clearly the requested information in both the paper itself, as well as the checklist and appendix attached.

**Ethics:**

One potential issue is that while informed consent was gathered during data collection from the household members, seeing as some households have no internet access, there may have been a mis/lack of understanding on the part of the households in how widely/in what way the images would be used. How the authors/team creating the data ensured that the participants understood and agreed to the entire extent of the data collection/use would be appreciated. Similarly, the authors mention that they allow people to request their images be removed from the dataset; but if the people are not aware that the dataset is out there in this format, used for certain purposes, or that they have this right, it becomes less meaningful.

The participants were not paid for their data/labor, which is also a point of concern - would the authors please elaborate on this choice?

**Relation To Prior Work:**

Similar datasets are described and compared; the uniqueness and contribution of the current dataset are made clear.

**Summary And Contributions:**

The authors present an image dataset of everyday household items from homes around the world, including metadata about the region, country, home monthly income, and tags of the objects in the images. The dataset includes images from homes with very low income and with no internet access, which is a subgroup that is not often represented in such dataset.

Having personally read about algorithm audits which use the web-based predecessor of this dataset (on GapMinder.org), I can confirm the work contributes greatly to the field, providing a dataset which allows for increased representation of vulnerable groups rarely represented in image datasets.

---

> ### Author Response · Authors · 2022-08-24
> **Added more details about the dataset and future uses**
>
> We thank the reviewer for their time and thoughtful feedback. We updated the paper as described below to address a number of the weaknesses they identified and clarify important details for future readers:
>
> - We added Section 6.3 to describe additional use cases of the dataset and clarify that it is better used for auditing and fine-tuning than training from scratch.
> - We extended the description of how the monthly income value is calculated into Section 4.2 based on Gapminder’s extensive description of how they quantify "income levels" on their website: https://www.gapminder.org/fw/income-levels/
> - We added more detail about how the data was collected in Section 3.3.
> - We updated Section 3.4 regarding the ethical concerns about the dataset.
>
> Regarding the ethical concerns, families were informed and needed to voluntarily agree to sign an informed conset document from the DollarStreet photo guide:
>
> https://docs.google.com/document/d/1ZNvVxul3blpzvl3VZQhkGmtKlC_RpiHHGF4ZygccC0g/edit#
>
> Moreover, the photos were collected by a global team of photographers, which Gapminder found by advertising on international network websites, receiving recommendations from friends/colleagues, and contacting some based on previous work. Both professional and volunteer photographers collected the photos, and some were compensated based on the fair market value of their country. As a result, monetary compensation for the families was avoided to ensure participation was completely voluntary and ensure photographers did not “buy” access to homes.
>
> Finally, we want to acknowledge the reviewer’s points about the list of topics and distribution of countries. While we agree that the boundary between topics is ambiguous and the regions are unequally represented, our goal with the paper is to release the dataset with as much information as possible and avoid introducing any biases in how the data is interpreted. The topic labels came directly from Gapminder. We took an initial stab at cleaning up the topics by fuzzy matching them to ImageNet classes, but this was not exhaustive. Defining a concise taxonomy of the topics is important and deserves dedicated effort, which is beyond the scope of this paper. We leave this as an excellent opportunity for future work.
>
> Similarly, due to our dependence on Gapminder, we are limited by what regions and countries the raw data contained. In the future, as Gapminder sources more raw images from different regions and countries, it is possible to improve the dataset. It is also possible to supplement the dataset with images from other sources outside of Gapminder.

---

### Review · Ethics_Reviewer_BfAU · 2022-08-22

**Recommendation:** 1

**Ethics Documentation:**

Largely in good order - but see above.

**Ethics Review:**

In many ways, this is an exemplary study. However, one of the core reviewers is correct when they note that:

"The participants were not paid for their data/labor, which is also a point of concern - would the authors please elaborate on this choice?"

In the checklist, the authors of the paper refer back to section 3.3. and 3.4 for details on these questions. However, having reviewed the sections I do not find these additional details provided. Although, this does not comprise a hard check on publication of the paper it is something that the authors should speak to if they want their paper to be best in class. We cannot simply assume that the actions undertaken by NGOs are non-exploitative. A fuller description of the program undertaken by Gapminder, even if only a few sentences, would certainly help assuage these concerns.

---

> ### Author Response · Authors · 2022-08-24
> **Families weren’t paid to ensure photographers did not “buy” access to homes.**
>
> We thank the reviewer for their time and thoughtful feedback. We updated Sections 3.3 and 3.4 with more detail about who was paid and why. Briefly, the photos were collected by a global team of photographers, which Gapminder found by advertising on international network websites, receiving recommendations from friends/colleagues, and contacting some based on previous work. Both professional and volunteer photographers collected the photos, and some were compensated based on the fair market value of their country. Monetary compensation for the families was avoided to ensure photographers did not “buy” access to homes and ensure participation was completely voluntary.

---

### Meta-Review · Area_Chair_yTT8 · 2022-09-10

**Recommendation:** Accept
**Confidence:** 2

**Metareview:**

The AC has taken into account all strengths and weaknesses mentioned by reviewers in making a recommendation for this paper. Here is the summary of what was considered in carefully considering a final decision:

* Reviewers agree that the presented resource, the Dollar Street dataset, addresses a need for models that can be held more accountable in terms of geographic diversity and demographic diversity.
* There is generally no issues in terms of methodology or soundness of the proposed resource.
* Unfortunately, experiments on demonstrating the usefulness of this resource are limited, and largely replicate what was already found in Devries et al. 2019, a workshop paper at CVPR from three years ago which also used the same source of data -- and at the same scale. The analysis in this previous paper in some way go further by matching image categories using human evaluators individually for each image -- as pointed out by reviewer dcyB.
* The dataset was downloaded from a third party's website and as such the authors had no control in how the images were collected, nor there was any effort to enhance this resource in terms of annotations, diversity or size.
* The dataset size is 38k images which is a rather small dataset for training or enhancing a model, and its use would be limited mostly as a benchmark dataset. As a benchmark however experiments seem limited as pointed by reviewer eAoC. An experiment on training on this dataset and evaluating on this dataset is performed only with a Resnet network, however it is hard to assess what is the significance of this experiment.

There are various suggestions made by reviewers below that would enhance the quality of this paper. Despite shortcomings, reviewers are still enthusiastic to see this dataset in an easy to use format as a benchmark. I also strongly suggest the authors to revise their section on licensing where it says "Training machine learning models on public domain work is widely accepted legally." as it does not seem appropriate for this paper to be making determinations or recommendations as to what is considered legal or not.

---

### Decision · Program_Chairs · 2022-09-16

Accept